# Chemical Modification as a Method of Improving Biocompatibility of Carbon Nonwovens

**DOI:** 10.3390/ma14123198

**Published:** 2021-06-10

**Authors:** Justyna Frączyk, Sylwia Magdziarz, Ewa Stodolak-Zych, Ewa Dzierzkowska, Dorota Puchowicz, Irena Kamińska, Małgorzata Giełdowska, Maciej Boguń

**Affiliations:** 1Institute of Organic Chemistry, Faculty of Chemistry, Lodz University of Technology, Zeromskiego 116, 90-924 Lodz, Poland; sylwia.magdziarz@dokt.p.lodz.pl; 2Department of Biomaterials and Composites, Faculty of Materials Science and Ceramics, AGH-UST University of Science and Technology, A. Mickiewicza 30, 30-059 Krakow, Poland; stodolak@agh.edu.pl (E.S.-Z.); dzierzkowska@agh.edu.pl (E.D.); 3Łukasiewicz Research Network-Textile Research Institute, Brzezińska 5/15, 92-103 Lodz, Poland; dorota.puchowicz@iw.lukasiewicz.gov.pl (D.P.); irena.kaminska@iw.lukasiewicz.gov.pl (I.K.); malgorzata.gieldowska@iw.lukasiewicz.gov.pl (M.G.)

**Keywords:** carbon nonwoven fabric, surface modification of carbon materials containing Csp^2^, aromatic amine, diazonium salt, incorporation of functional groups onto the material surface, RGD- peptide, surface free energy, cells adhesion

## Abstract

It was shown that carbon nonwoven fabrics obtained from polyacrylonitrile fibers (PAN) by thermal conversion may be modified on the surface in order to improve their biological compatibility and cellular response, which is particularly important in the regeneration of bone or cartilage tissue. Surface functionalization of carbon nonwovens containing C–C double bonds was carried out using in situ generated diazonium salts derived from aromatic amines containing both electron-acceptor and electron-donor substituents. It was shown that the modification method characteristic for materials containing aromatic structures may be successfully applied to the functionalization of carbon materials. The effectiveness of the surface modification of carbon nonwoven fabrics was confirmed by the FTIR method using an ATR device. The proposed approach allows the incorporation of various functional groups on the nonwovens’ surface, which affects the morphology of fibers as well as their physicochemical properties (wettability). The introduction of a carboxyl group on the surface of nonwoven fabrics, in a reaction with 4-aminobenzoic acid, became a starting point for further modifications necessary for the attachment of RGD-type peptides facilitating cell adhesion to the surface of materials. The surface modification reduced the wettability (*θ*) of the carbon nonwoven by about 50%. The surface free energy (SFE) in the chemically modified and reference nonwovens remained similar, with the surface modification causing an increase in the polar component (ɣ_p_). The modification of the fiber surface was heterogeneous in nature; however, it provided an attractive site of cell–materials interaction by contacting them to the fiber surface, which supports the adhesion process.

## 1. Introduction

One of the directions of current medical development is the search for new materials that through their functionality are able to replace damaged tissues or support their reconstruction process. Among the materials that still remain in the area of interest of many scientific teams are carbon fibers and fibrous structures based on them [1]. The process of the preparation of carbon fibers enables modification of the fiber structure as early as the stage of precursor fiber formation. As was shown in a study by Mikołajczyk et al. [2], better results of osteointegration with bone tissue were achieved if hydroxyapatite (HAp) or tricalcium phosphate (TCP) was introduced into the polymer precursor (PAN). Frączek et al. [3] showed an increase in bioactive efficiency when wollastonite was used. Moreover, as was shown in other studies [4,5,6], it is possible to modify the surface of carbon fibers with various additives and polymers and to obtain composites with beneficial biological properties on this basis.

A new direction of surface modification of medical carbon fibers (thus nonwoven fabrics) is their activation by controlled chemical treatment and introduction onto the surface of some compounds, affecting the chemical, physical, and biochemical properties of carbon fibers. A widely used reaction for the functionalization of carbon-based materials (including fullerenes, carbon nanotubes, and nanodiamond) is oxidation, either with classical oxidants [7,8,9] or plasma [10,11,12]. In case of carbon nonwoven fabric, the extensive use of oxidation as a method of functionalization or prefunctionalization is limited due to the fact that the material is not very reactive, resulting in the use of drastic oxidation conditions (strong acids), while on the other hand, the reaction conditions have to be mild enough to avoid destruction of both the structure and the fibrous form of the modified material.

Mild surface functionalization conditions of carbon-based materials are provided by using a method based on the reaction between diazonium salts and carbon-based materials containing carbon atoms with sp^2^ hybridization [12,13,14,15,16], such as carbon nanotubes and fullerenes. A method using diazonium salts has also been successfully used for the orthogonal functionalization of olefinic nanodiamonds formed by a laser treatment process [17]. In addition, a number of methods are known for the preparation of diazonium salts [18,19], and a number of aromatic amine derivatives are available, which allows the introduction of different functional groups onto the surface of the material. The functionalization method of using diazonium salts of aromatic amine derivatives has not been used so far to modify fibers or carbon nonwovens derived from a polymer precursor (PAN).

The aim of such surface functionalization of materials, including carbon-based materials, is to increase their biological potential. In this case, it is about improving the cell adhesion process and, consequently, cell proliferation. Increasing the specific interaction between the material and cells is achieved by modifying the materials with proteins, polypeptides, or peptides. Peptides containing the RGD (Arg-Gly-Asp) motif are often used to obtain materials useful in various medical fields [20,21,22,23,24]. This is due to the fact that this sequence stimulates cell adhesion. RGD is a structural motif in many adhesive ECM proteins (for instance fibronectin, fibrinogen, vitronectin, osteopontin, and some others) of living organisms, and it activates the adhesion of more than one cell type by acting on more than one type of adhesion receptor, which in turn improves cell adhesion and cell survival [25,26,27]. In multicellular organisms, the interaction of cells with surrounding cells and the ECM (extracellular matrix) is mediated by cell adhesion receptors. Among these receptors, differentiated integrins play a major role. Integrins are a highly diverse class of adhesion receptors that play a significant role in the function of all higher organisms [28,29]. They are not only responsible for binding compounds, but are also crucial in many processes: embryogenesis, cell differentiation, immune response, wound healing processes, and hemostasis [30,31,32,33,34,35,36].

The process of cell adhesion has not been precisely understood. On the one hand, the adhesion of prokaryotic (bacteria) and eukaryotic (mammalian) cells strongly depends on their structure and the (in)presence of a cell wall, but on the other hand, surface free energy plays a key role [37,38]. New results show that surface chemistry strongly influences the value of polar and dispersive components of surface energy [39]. Thus, the adhesion of eukaryote cells increases with surface free energy, with the polar component playing a major role, while the adhesion rate of bacterial (prokaryotic) cells increases with surface hydrophobicity, thus revealing that hydrophobic forces drive adhesion kinetics [40].

The aim of this study was to investigate the possibility of chemical modification of the surface of carbon nonwoven fabric made of micrometer-sized carbon fibers, using a method based on in situ formation of diazonium salts of aromatic amine derivatives. The carbon atoms with sp^2^ hybridization that are present on the surface of carbon nonwovens react with diazonium salts similarly to classical carbon nanomaterials containing unsaturated bonds. The advantage of this approach is the preparation of modified materials in which the modifying agent is bound with the surface via a stable C–C bond. The possibility of using this method of functionalization allows the incorporation of derivatives containing various functional groups of electron-acceptor or electron-donor character on the surface of carbon nonwoven fabrics. The proposed approach significantly affects the physicochemical properties of carbon nonwoven substrates and can enable further functionalization of carbon nonwovens, e.g., with biologically active RGD-type peptides. This approach determines the interaction with integrins, receptors responsible for cell adhesion process. As part of the experimental work, we developed a method for the chemical modification of the surface of carbon nonwoven fabrics derived from a polymeric precursor (PAN) with a group of structurally diverse aniline derivatives, i.e., compounds forming diazonium salts in situ. The material formed under the treatment with 4-aminobenzoic acid in the second stage of modification was used for the incorporation of RGD peptide onto the surface of carbon nonwoven fabric. This stage was implemented by using ethylenediamine as a linker. 4-(4,6-Dimethoxy-1,3,5-triazin-2-yl)-4-methylmorpholinium toluene-4-sulfonate (DMT/NMM/TosO^−^) was used as a coupling reagent to activate the carboxylic function. The effectiveness of the modification was examined each time by spectroscopic (FTIR), physicochemical (wettability), and microstructural (SEM) methods.

## 2. Materials and Methods

### 2.1. Preparation and Characterization of Carbon Nonwovens

For this study, needled nonwovens made of polyacrylonitrile (PAN) fibers with weight of approximately 300 g/m^2^ were used. The nonwovens were treated with a two-step thermal treatment including thermal stabilization (oxidation) and carbonization. At the thermal stabilization stage, the optimum properties of the nonwovens were obtained using an oxidizing atmosphere (air) and a time/temperature sequence: 6 h/150 °C followed by 6 h/240 °C. The obtained intermediate product (oxyPAN) was subjected to a carbonization process in the next stage. A protective atmosphere (argon) was used, and the nonwoven fabrics were kept for 10 min at 1000 °C. During the thermal treatment, nonwovens changed their color: from white (typical for PAN precursor) to red-brown (oxyPAN) to black (characteristic for carbon fibers) (CF).

The surface densities of PAN, oxyPAN, and CF nonwovens were determined by hydrostatic weighing. Apparent density was calculated based on Equation (1).
(1)$$\mathrm{Apparent}\;\mathrm{density}\;\mathrm{of}\;\mathrm{scaffold}=\frac{\mathrm{mass}\;\mathrm{of}\;\mathrm{the}\;\mathrm{sample}\;\left(g\right)}{\mathrm{sample}\;\mathrm{thickness}\;\left(\mathrm{cm}\right)\times \mathrm{sample}\;\mathrm{area}\left({\mathrm{cm}}^{2}\right)}$$

The porosity of the nonwoven scaffolds was estimated using the gravimetric method Equation (2) [41].
(2)$$\mathrm{Porosity}\;\left(\%\right)=\left(1-\left(\frac{\mathrm{Apparent}\;\mathrm{density}\;\mathrm{of}\;\mathrm{scaffold}\;\left(\frac{g}{{\mathrm{cm}}^{3}}\right)}{\mathrm{Average}\;\mathrm{bulk}\;\mathrm{density}\;\mathrm{of}\;\mathrm{the}\;\mathrm{polymers}\left(\frac{g}{{\mathrm{cm}}^{3}}\right)}\right)\right)$$

Wettability tests were conducted on a DSA 25 goniometer, Kruss (Kruss, Hamburg, Germany), using UHQ water as a measuring liquid. A sessile drop of a deionized water (0.15–0.25 μL in volume) was deposited at the center of the nonwoven surface. The measurements were performed at room temperature for polymer precursor (PAN) and samples after thermal conversion. Prior to the test, nonwoven samples after chemical modification were air-dried (placed on a wet support for 24 h at 25 °C). The results of five measurements for each sample were averaged and presented as mean values with a significance level of α = 0.05.

Surface free energy was calculated based on a mathematical analysis of direct measurements of the contact angle (as described above) for deionized water and diiodomethane (nonpolar liquid, Avantor, Gliwice, Poland). The component method was applied (dispersive and polar component Owens–Wendt method) with the system of equations as follows:(3)$$\{\begin{array}{c} \sqrt{\gamma_{L}^{d}}\;+1.53\;\sqrt{\gamma_{S}^{d}\;\;}=7.8\;\cdot \left(1+\cos\theta_{1}\right)\\ \sqrt{\gamma_{L}^{d}}\;+0.22\;\sqrt{\gamma_{S}^{d}\;\;}=3.65\;\cdot (1+\cos\theta_{2}) \end{array}$$
where *θ*_1_ represents the contact angle when wetting the material with water and *θ*_2_ represents the contact angle when wetting the material with diiodomethane.

The structures of PAN, oxyPAN, and CF nonwovens, as well as that of chemically modified nonwovens, were investigated by the FTIR method on FTIR spectrometer VERTEX 70 (Bruker, Bremen, Germany) using an ATR Golden Gate adapter. Resolution of 2 cm^−1^, 64 scans in the range 600–4000 cm^−1^. The results were recorded in the program BRUKER OPUS 6.5 (Version 6.5, Bruker, Kennewick, WA, USA).

For the chemical modification, CF carbon nonwovens, characterized by the presence of carbon atoms with sp^2^ hybridization on the surface, were used.

Measurements of the surface topography (SEM analysis) of the materials were carried out on a VEGA3 TESCAN (Tescan Osay Holding, Brno, Czech Republic). 

### 2.2. General Procedure for the Functionalization of Carbon Nonwoven Fabric by Using In Situ Generated Diazonium Salts

A carbon nonwoven (**CF**) of size 10 × 10 mm^2^ was wetted by treating with 15 mL DMF (J.T. Baker Avantor Performance Materials, Gliwice, Poland) for 5 min. Then the nonwoven was treated with a mixture of DMF: 1 M HCl (15 mL) (1:1) for 5 min and 1 M HCl (15 mL) for 10 min. Next, nonwoven fabric was transferred to a round-bottom flask and treated with 25 mL of 1 M HCl and heated to 80 °C. The reaction mixture was stirred gently. After reaching the required temperature, isoamyl nitrite (0.7 mL, 5 mmol) (Sigma-Aldrich, Poznan, Poland) and aniline derivative (5 mmol) were added to the solution. The reaction was conducted at 80 °C for 18 h. Finally, the nonwoven was washed with DMF (3 × 5 mL) and DCM (3 × 5 mL).

#### 2.2.1. Functionalization of **CF** with 4-Bromoaniline, Preparation of **CF-1a**

The prewetted carbon nonwoven (10 × 10 mm^2^) was placed in a round-bottom flask and treated with 25 mL of 1 M HCl, and then the mixture was heated to 80 °C. After reaching the required temperature, isoamyl nitrite (0.7 mL, 5 mmol) and 4-bromoaniline (0.86 g, 5 mmol) (Sigma-Aldrich, Poznan, Poland) were added to the mixture. The reaction was carried out at 80 °C for 18 h. Finally, the nonwoven fabric was washed with DMF (3 × 5 mL) and DCM (3 × 5 mL).

#### 2.2.2. Functionalization of **CF** with 4-Chloroaniline, Preparation of **CF-1b**

The following materials were used for the reaction: 10 × 10 mm^2^ prewetted carbon nonwoven fabric, 25 mL of 1 M HCl, isoamyl nitrite (0.7 mL, 5 mmol), and 4-chloroaniline (0.64 g, 5 mmol) (Sigma-Aldrich, Poznan, Poland). The reaction was carried out at 80 °C for 18 h. The modified **CF-1b** nonwoven was washed with DMF (3 × 5 mL) and DCM (3 × 5 mL).

#### 2.2.3. Functionalization of **CF** with 4-Aminobenzoic Acid, Preparation of **CF-1c**

The following materials were used for the reaction: 10 × 10 mm^2^ prewetted carbon nonwoven fabric, 25 mL of 1 M HCl, isoamyl nitrite (0.7 mL, 5 mmol), and 4-aminobenzoic acid (0.71 g, 5 mmol) (Sigma-Aldrich, Poznan, Poland). The reaction was carried out at 80 °C for 18 h. The modified **CF-1c** nonwoven was washed with DMF (3 × 5 mL) and DCM (3 × 5 mL).

#### 2.2.4. Functionalization of **CF** with 1,4-Diaminobenzene, Preparation of **CF-1d**

The following materials were used for the reaction: 10 × 10 mm^2^ prewetted carbon nonwoven fabric, 25 mL of 1 M HCl, isoamyl nitrite (0.7 mL, 5 mmol), and 1,4-diaminobenzene (0.54 g, 5 mmol) (Sigma-Aldrich, Poznan, Poland). The reaction was carried out at 80 °C for 18 h. The modified **CF-1d** nonwoven was washed with DMF (3 × 5 mL) and DCM (3 × 5 mL).

#### 2.2.5. Functionalization of **CF** with 4-Aminophenol, Preparation of **CF-1e**

The following materials were used for the reaction: 10 × 10 mm^2^ prewetted carbon nonwoven fabric, 25 mL of 1 M HCl, isoamyl nitrite (0.7 mL, 5 mmol), and 4-aminophenol (0.55 g, 5 mmol) (Sigma-Aldrich, Poznan, Poland). The reaction was carried out at 80 °C for 18 h. The modified **CF-1e** nonwoven was washed with DMF (3 × 5 mL) and DCM (3 × 5 mL).

### 2.3. Attachment of Ethylenediamine to the Carbon Nonwoven Containing a Free Carboxyl Group on the Surface (**CF-1c**). Synthesis of Modified Nonwoven **CF-2**

A functionalized carbon nonwoven (**CF-1c**) containing carboxyl groups on the surface was used to incorporate diamine derivative. The carbon nonwoven **CF-1c** was treated with a solution containing 4-(4,6-dimethoxy-1,3,5-triazin-2-yl)-4-morpholine (DMT/NMM/TosO^−^) [42] (0.21 g, 5 mmol) and *N*-methylmorpholine (NMM) (Sigma-Aldrich, Poznan, Poland) (550 µL, 5 mmol) in DMF (5 mL) as a solvent. The reaction of carboxyl group activation was carried out for 90 min at room temperature. Then the excess reactants were removed, and the nonwoven fabric was washed twice with DCM (5 mL) and treated with ethylenediamine (Sigma-Aldrich, Poznan, Poland) (0.45 mL, 7 mmol) in DCM. The condensation reaction was carried out for 12 h at room temperature. After this time, the nonwoven was washed with DMF (3 × 5 mL) and DCM (3 × 5 mL). The modification product was a nonwoven CF-C6H_4_-CONH-(CH_2_)_2_-NH_2_ (**CF-2**) containing free amine groups on the surface.

### 2.4. Incorporation of a Peptide Containing an RGD Motif to the Surface of a Carbon Nonwoven Fabric CF-C_6_H_4_-CONH-(CH_2_)_2_-NH_2_ (**CF-2**). SYNTHESIS of CF-C_6_H_4_-CONH-(CH_2_)_2_-NHCO-DGR-Ac (**CF-3**)

The nonwoven fabric containing primary amine groups on the surface (**CF-2**) was further functionalized with Ac-RGD-OH peptide. In the first stage, Ac-RGD-OH peptide (160 mg, 0.5 mmol) was treated with DMT/NMM/TosO^−^ (0.31 g, 0.75 mmol) and NMM (165 µL, 1.5 mmol) in DMF (3 mL). Then the functionalized carbon nonwoven fabric **CF-2** was treated with an acylation solution. The condensation reaction was carried out for 12 h at room temperature. At the end of the process, the nonwoven fabric was washed with DMF (3 × 5 mL), a mixture of water–DMF (1:1) (3 × 5 mL), and DCM (3 × 5 mL) and dried in a vacuum desiccator to constant weight.

### 2.5. Synthesis of Ac-RGD-OH

In the first stage of the synthesis, Fmoc-Asp(OtBu)-OH (0.617 g, 1.5 mmol) was attached to 1 g of 2-chlorotrityl resin according to the procedure described in the Appendix A. In the reaction, DIPEA (522 µL, 3 mmol) was used. Resin loading was determined according to the spectrophotometric general procedure described in the Appendix A. The calculated resin loading was 0.4 mmol/g. Afterwards, deprotection of the Fmoc group was performed on the resin containing the C-terminal amino acid according to the general procedure described in the Appendix A. For the synthesis of the final peptide, Fmoc-Gly-OH (0.357 g, 1.2 mmol), Fmoc-Arg(Pbf)-OH (0.209 g, 1.2 mmol) were used. For amino acid incorporation, DMT/NMM/TosO^−^ (0.496 g, 1.2 mmol) and NMM (660 µL, 3.6 mmol) were used. Reactions were carried out according to the general procedure described in the Appendix A. The removal of the Fmoc protecting group was performed according to the general procedure described in the Supplementary Material using 25% piperidine solution in DMF. For the incorporation of the arginine residue, removal of the Fmoc group was applied using a 2% DBU solution. The final stage of the reaction involved the acylation reaction of the amino group of the arginine residue. For this reaction, 10 mmol acetic anhydride (1.02 mL) and 25 mmol DIPEA (3.23 mL) were used. The reaction was carried out for 1 h. The final product was isolated after completing the reaction according to the general procedure described in the Appendix A. The final product was obtained with a purity of 90%. Its structure was confirmed by MS, *m*/*z* = 389.1921 g/mol, which corresponds to [M + H]^+^ of the expected product with M = 388.37 g/mol (see Appendix A).

## 3. Results and Discussion

The nonwoven fabric made out of PAN precursor fibers obtained by wet solution spinning was treated with a two-step thermal treatment involving thermal stabilization (oxidation) and carbonization. This treatment led to the contraction of both the nonwoven and the single fiber. The diameter of the oxidized fiber (oxyPAN) was about 30% lower than that of the PAN fiber. The contraction of the carbonized fiber with regard to the oxidized one was marginal at about 2–3% (Table 1).

Based on the results of porosity tests, the nonwovens PAN, oxyPAN, and CF were found to have high open porosity (80–90%), which increased with subsequent stages of thermal treatment (Table 1). The contraction of fibers generates the formation of larger spaces between fibers, which are responsible for the porosity of the material (nonwovens). Thermal treatment itself significantly influences the physicochemical properties of fibers, demonstrated by the observed decrease of nonwoven wettability. It is known that the physicochemical properties of fibers (nonwovens), including wettability, allow the modulation of cell adhesion, a key parameter affecting the interaction of cells with the material, the ability to proliferate and grow, and thus the use of fibers/nonwovens as substrates for the preparation of scaffolds useful in regenerative medicine [37,43,44,45,46,47]. The highest wettability (hydrophilicity) was characteristic for PAN precursor nonwovens, while the most hydrophobic were carbon nonwovens (CF). As a result, the retention of carbon nonwovens decreased (Table 1).

The process of structural changes of PAN, oxyPAN, and CF fibers was observed by FTIR spectroscopy. Oxidation (stabilization) is one of the main stages of preparing nonwovens for the carbonization process. Analysis of the structure and mechanical properties of precursor fibers and carbon fibers was presented in our previous work [3], which included changing the average values of modified fibers after oxidation and carbonization, supramolecular structure, and microstructure fibers. At this stage, cyclization and dehydrogenation of polyacrylonitrile takes place, leading to the formation of cyclic structures in the polymer chain. The progress of the oxidation process of the PAN nonwoven was evidenced by the disappearance of the bands at wavelengths 2240 cm^−1^ and 1730 cm^−1^. These bands are associated with the presence of C≡N and C=O groups, respectively. The intensification of the band at wavelength 1600 cm^−1^ associated with C=C or C=N valence vibrations is also required (Figure 1).

The second stage of the carbon fiber synthesis was carbonization carried out at 1000 °C. The strong background given by carbon made it difficult to analyze the spectrum, but at the same time, it was evidence of an effective carbonization process.

CF carbon nonwovens characterized by the presence of sp^2^ carbon atoms on the surface [48] were used for chemical modification. A method based on using in situ generated diazonium salts was used to functionalize the surface of the nonwovens, which allowed the formation of materials in which the incorporated element is attached via a stable C–C covalent bond. In this study, there were experiments to functionalize carbon nonwoven fabric using diazonium salts obtained in situ from aromatic amines and isoamyl nitrite (III). In the reaction carried out under the influence of hydrochloric acid (HCl), the nitrosonium ion NO^+^ is generated (Figure 2).

Then the primary amine group of the aromatic amine derivative forms a diazonium cation, which is transformed into a benzoic cation (Figure 3a) that reacts with sp^2^ hybridization carbon atoms (Figure 3b).

Substituted aniline derivatives containing both electron-acceptor (COOH, Cl, Br) and electron-donor groups (OH, NH_2_) were used for the functionalization of carbon nonwoven fabric **CF**. Using a procedure involving pretreatment of the carbon nonwoven with DMF (a mixture of DMF-1 M HCl and 1 M HCl, organic solvents), reactions were carried out with in situ generated diazonium salts formed in the presence of isoamyl nitrite [17]. Finally, **CF-1a**–**CF-1e** carbon nonwovens were obtained. In order to confirm the incorporation of benzene derivatives with various functional groups onto the surface of the carbon nonwoven fabric, FTIR studies were performed (Figure 4).

Based on the FTIR results, for none of the modified nonwovens, after the incorporation of the substituted aromatic amine derived residue, was the spectrum identical to that of the initial carbon nonwoven **CF** (Figure 4). For the nonwoven **CF-1a** (Figure 4b), a band was found in the range 1600–1500 cm^−1^ corresponding to the C=C vibration in the aromatic ring, while the band found in the range 3070–3030 cm^−1^ corresponds to the C-H vibrations of the aromatic ring. In case of the nonwoven **CF-1b** (Figure 4c), the following were detected: a peak at about 1100 cm^−1^ corresponding to C-H vibrations of the aromatic ring, a peak in the range 1600–1500 cm^−1^ corresponding to C=C vibrations in the aromatic ring, and a band at 3070–3030 cm^−1^ corresponding to C-H vibrations in the aromatic ring. For the nonwoven **CF-1c** (Figure 4d), there was detected a peak at the wavelength 1630 cm^−1^ from valence C=C vibrations and a peak in the range 2961–3265 cm^−1^ from vibrations of the -OH group (the characteristic range of vibrations for this band is 3600–3300 cm^−1^). Moreover, the peak around the wavelength 1650 cm^−1^ corresponds to the vibrations of the C=O carbonyl group of carboxylic acids. Also observed were the characteristic band around 1100 cm^−1^ corresponding to C-H vibrations in the aromatic ring and the peak in the range 1600–1500 cm^−1^ corresponding to C=C vibrations in the aromatic ring. For the nonwoven **CF-1d** (Figure 4e) was detected a peak around the wavelength 1100 cm^−1^ corresponding to C-H vibrations in the aromatic ring, a band in the range 1600–1500 cm^−1^ characteristic for C=C vibrations in the aromatic ring, a peak around the wavelength 1300 cm^−1^ corresponding to C-N vibrations of the amines, and a band around the wavelength 3300 cm^−1^ associated with vibrations in the NH_2_ group of amines. In case of the nonwoven **CF-1e** (Figure 4f), there was detected a peak around 1100 cm^−1^ corresponding to C-H vibrations of the aromatic ring and a band in the range 1600–1500 cm^−1^ corresponding to C=C vibrations an aromatic molecule. However, the most characteristic band confirming modification efficiency was the band at 3600 cm^−1^ corresponding to the vibration band in the -OH group of phenols.

For some modified materials, microscopic investigations using scanning electron microscopy (SEM) were also performed (Figure 5).

By analyzing the images presented above (Figure 5), it was found that the most changed morphology of the surface-modified nonwoven fabric was for material functionalized with 4-aminobenzoic acid (**CF-1c**) (Figure 5d). However, for the other modified nonwovens, it was observed that their morphology changed compared to the unmodified nonwoven **CF** (Figure 5a).

The incorporation of a variety of functional groups onto the surface of carbon nonwoven fabric, through coupling reactions of in situ generated diazonium salts derived from aromatic amines, enables their use in modifications that allow the modulation of biological properties. In this study, carbon nonwoven fabric modified with 4-aminobenzoic acid (**CF-1c**) was used as a substrate for the incorporation of RGD peptide, which should provide a positive effect on improving the functionality of carbon fiber-based materials in terms of bone and cartilage tissue regeneration. The presence of a carboxyl group on the surface of the carbon nonwoven fabric made it possible to obtain a conjugate with the RGD peptide bound by an amide bond (Figure 6). The first stage of modification was the incorporation of an ethylenediamine residue to the carboxyl group on the nonwoven material **CF-1c**. As a coupling reagent to effective activation of the carboxyl function on the surface of solid materials [49,50,51], 4-(4,6-dimethoxy-1,3,5-triazin-2-yl)-4-methylmorpholiniumtoluene-4-sulfonate (DMT/NMM/TosO^−^) [52] was used. As a result of the reaction with ethylenediamine, there was obtained a carbon nonwoven fabric **CF-2** containing a primary amine group on the surface, which was used for coupling with Ac-RGD-OH peptide, previously activated with DMT/NMM/TosO^−^. The final product of the reaction was a carbon nonwoven **CF-3**, containing peptide on the surface (Figure 6). The use of ethylenediamine and Ac-RGD-OH peptide for the modification of the nonwoven CF-1c led to obtaining the material **CF-3**, which has a significantly higher biocompatibility due to the presence of a peptide fragment and a linker (H_2_N-CH_2_CH_2_-NH_2_) connecting the carbon material with the biologically active ligand (Ac-RGD-OH) via an amide bond. Furthermore, **CF-3** should also have an effect on the increased stability of the modified material because of the fact that the amide bond is more stable compared to the ester bond. The use of Ac-RGD-OH as a biologically active ligand should also improve the stability of the material because the presence of an acyl group on the *N*-terminal amino acid of the peptide eliminates or significantly reduces the proteolytic activity of aminopeptidases. The effect of carboxypeptidases on the material **CF-3** should also be limited due to the fact that the carboxyl group of the C-terminal amino acid is linked by an amide bond to the linker molecule.

The efficiency of the modification of the nonwoven **CF-1c** with ethylenediamine and Ac-RGD-OH peptide was investigated by the FTIR method (Figure 7).

Results of the spectrophotometric studies confirmed the chemical functionalization of the surface of the carbon nonwoven. FTIR spectra confirmed the presence of characteristic bands for specific functional groups (Figure 7b). On the FTIR spectrum of the nonwoven **CF-3**, obtained as a result of functionalization with Ac-RGD peptide and diamine linker, can be observed a peak around 1700 cm^−1^ characteristic for the carbonyl group C=O of amides and bands in the range 2800–2900 cm^−1^ corresponding to stretching CH vibrations (Figure 7b). Additional bands confirming the efficiency of incorporation of the peptide fragment onto the nonwoven surface are a peak around 700 cm^−1^ corresponding to the vibrations of the aromatic ring and a peak in the range 1500–1300 cm^−1^ characteristic for C–C bonds. In addition, on the spectrum recorded for the nonwoven **CF-2**, the presence of characteristic bands confirms the efficiency of ethylenediamine incorporation to the carboxyl group (Figure 7a). These bands are: a peak in the range 1600–1500 cm^−1^ corresponding to C=C vibrations in the aromatic ring, a peak around 1700 cm^−1^ characteristic for the C=O group of carboxylic acids derivatives, and a peak around 3500 cm^−1^ corresponding to the vibrations in the NH_2_ group.

Based on SEM studies (Figure 8) of the nonwoven fabric modified with ethylenediamine (**CF-2**) and of that modified with Ac-RGD-OH peptide (**CF-3**), it was observed that an additional layer was formed after the surface functionalization compared to the unmodified nonwoven fabric.

The visible changes in the morphology of the fiber surface may be a result of the ability of the nonwoven **CF-3** containing the polar Ac-RGD peptide on the surface to interact with water, thereby forming structures that are more likely to interact with polar cellular components. The introduction of polar groups or peptides on the surface of hydrophobic carbon material should affect cell adhesion due to a change in the polarity of the material’s surface and thus enable modulation of the material’s ability to interact with cells.

The physicochemical studies carried out show that the initial carbon fiber is characterized by an almost hydrophobic surface (~90°), with the dispersion component of the surface energy being the dominant one. With reference to published data [39,40], it appears that such a surface would be attractive to prokaryotic cells, particularly gram-negative bacteria (e.g., *E. coli*). Chemical modification by introduction of polar groups (i.e., COOH, CO, CONH) (Figure 9) on the surface of the nonwovens leads to an increase in hydrophilicity of the material (ca. 70°, Table 2). In both the modified nonwoven fabrics **CF-1c** and **CF-3**, where polar groups are present, the polar component (ɣ_p_) dominates in the surface energy, and its value increases three times in comparison with the dispersion component (ɣ_p_). The trawl surface does not have a homogeneous character as evidenced by the standard deviation values. This means that the introduction of polar groups or peptides on the surface of hydrophobic carbon material should affect mammalian cell adhesion due to a change in the polarity of the material’s surface, and thus enable modulation of the material’s ability to interact with cells [39].

It is worth noting that the chemical treatment did not change the morphology of the fiber surface, which remained smooth and maintained both shape and size (diameters). It can be assumed that the preservation of the existing fibrous microstructure with altered fiber surface chemistry may facilitate adhesion. The domain-like nature of the surface should on the one hand stimulate mammalian cells to adhere and produce an adhesion plate; on the other hand, the presence of hydrophobic groups (CH_2_/CH_3_) may drive the kinetics of adhesion to the surface [40].

The results of the conducted research allowed us to demonstrate that the materials modified in this way are characterized by significantly higher wettability compared to unmodified materials (see Appendix A, Appendix A).

This property should therefore influence better cell adhesion to the modified carbon nonwovens. The study of the effect of modified carbon nonwoven fabrics on cell proliferation ability, as well as the evaluation of their other biological properties, will be the subject of the next stages of this research.

## 4. Conclusions

The study has shown that it is possible to functionalize the surface of carbon nonwovens derived from a polymer precursor (PAN) containing a C=C double bond using in situ generated diazonium salts derived from aromatic amines. It was found that it is possible to modify the surface of carbon nonwoven fabric using in situ generated diazonium salts derived from aromatic amines containing both electron-acceptor and electron-donor substituents. The efficiency of surface modification of carbon nonwovens was confirmed by the FTIR method. Aniline derivatives containing electron-donor substituents were found to be better substrates for coupling to the C=C double bond, which may be related to the fact that reactive diazonium salt obtained from anilines containing electron-donor substituents (OH or NH_2_) may undergo autocoupling, forming derivatives that are removed during the removal of excess the reactants. A possible side reaction may reduce the amount of diazonium salt reacting with the surface of the carbon nonwoven, resulting in a lower degree of modification. It has been shown that the developed functionalization method not only allows for the incorporation of various functional groups on the material surface, but also affects the morphology of fibers in the nonwoven as well as physical properties (hydrophilicity). The use of 4-aminobenzoic acid for the modification of carbon nonwoven fabric enabled obtaining a material containing carboxyl groups on the surface, which can be further functionalized using methods typical for peptide incorporation on the surface of materials. For this purpose, a diamine linker (ethylenediamine) was coupled to the carboxylic function, which allowed effective incorporation of Ac-RGD peptide into the carbon nonwoven fabric. The Ac-RGD peptide is a ligand for integrin receptors, which play a key role in cell proliferation and differentiation. Studies on the influence of modified carbon nonwovens on cell proliferation ability, as well as the evaluation of their biological properties, will be the subject of the next stages of this research.

## Figures and Tables

**Figure 1 materials-14-03198-f001:**
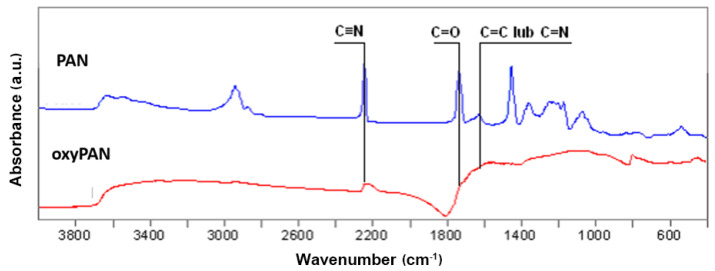
FTIR spectrum of PAN precursor nonwoven fabric and nonwoven fabric after oxidation process (oxyPAN).

**Figure 2 materials-14-03198-f002:**
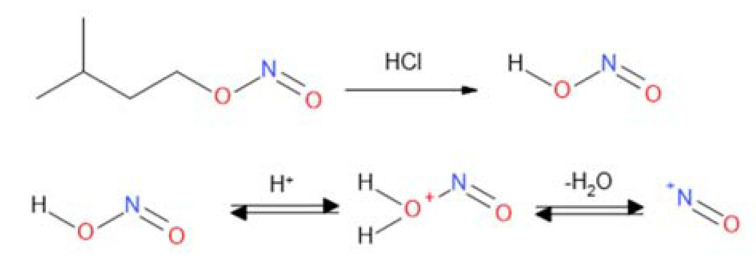
Generation of the nitrosonium ion.

**Figure 3 materials-14-03198-f003:**
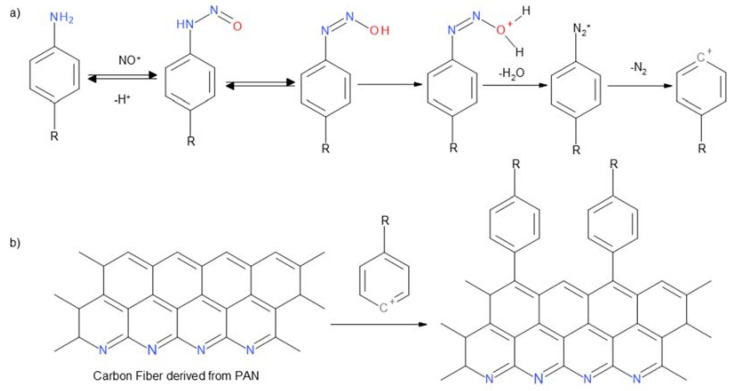
(**a**) Benzoic cation generation reaction; (**b**) coupling reaction to the surface of materials containing carbon atoms with Csp^2^ configuration.

**Figure 4 materials-14-03198-f004:**
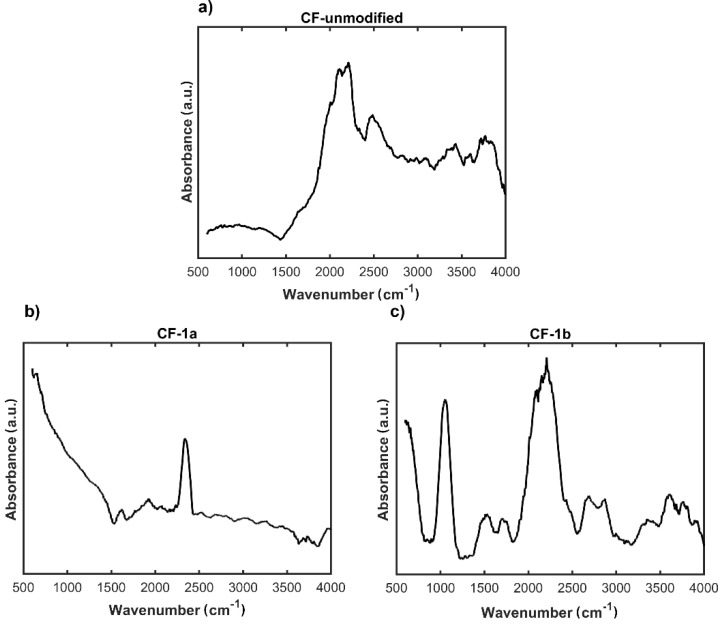
FTIR spectra of: (**a**) unmodified carbon nonwoven (**CF**); (**b**) carbon nonwoven fabric modified with 4-bromoaniline (**CF-1a**); (**c**) carbon nonwoven fabric modified with 4-chloroaniline (**CF-1b**); (**d**) carbon nonwoven fabric modified with 4-aminobenzoic acid (**CF-1c**); (**e**) carbon nonwoven fabric modified with 1,4-diaminobenzene (**CF-1d**); (**f**) carbon nonwoven fabric modified with 4-aminophenol (**CF-1e**).

**Figure 5 materials-14-03198-f005:**
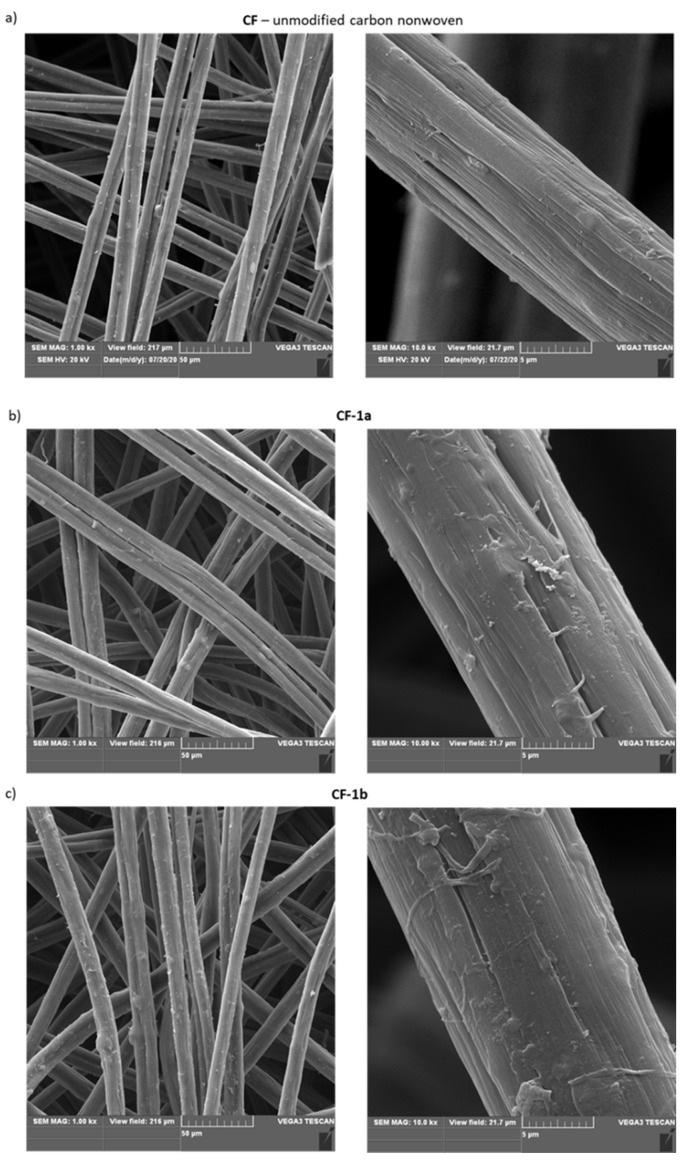
SEM images of: (**a**) unmodified carbon nonwoven (**CF**); (**b**) carbon nonwoven modified with 4-bromoaniline (**CF-1a**); (**c**) carbon nonwoven modified with 4-chloroaniline (**CF-1b**); (**d**) carbon nonwoven modified with 4-aminobenzoic acid (**CF-1c**); (**e**) carbon nonwoven modified with 1,4-diaminobenzene (**CF-1d**); (**f**) carbon nonwoven modified with 4-aminophenol (**CF-1e**).

**Figure 6 materials-14-03198-f006:**
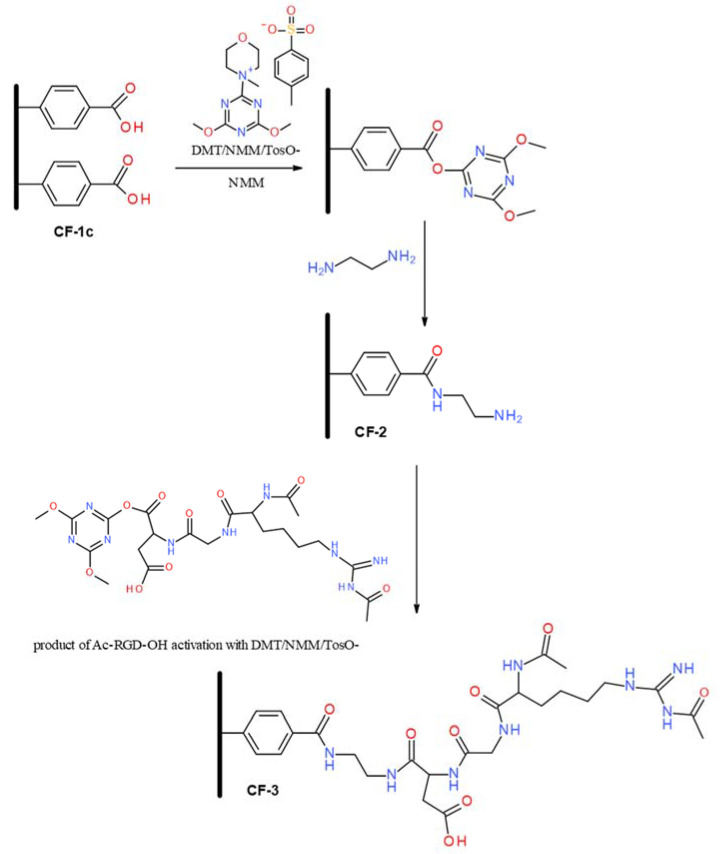
Scheme of the modification of the nonwoven **CF-1c** with ethylenediamine and Ac-RGD-OH peptide.

**Figure 7 materials-14-03198-f007:**
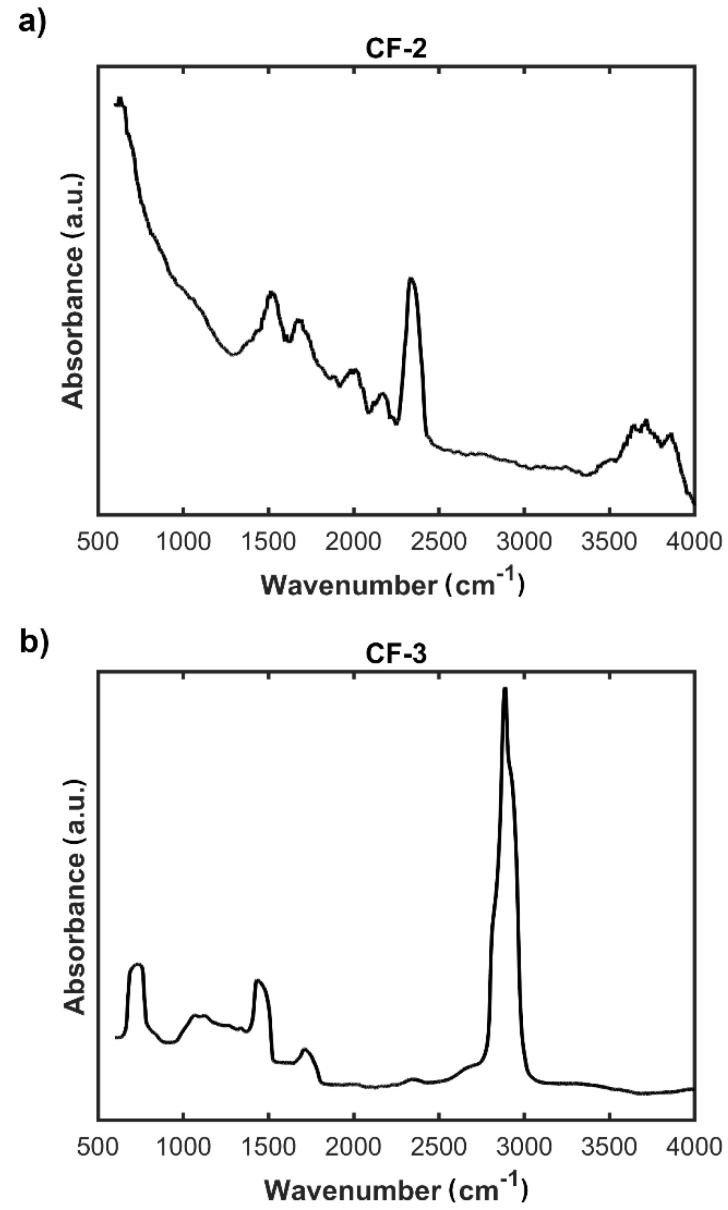
FTIR spectra of: (**a**) modified carbon nonwoven fabric containing C_6_H_4_-CONH-CH_2_CH_2_-NH_2_ on the surface (**CF-2**); (**b**) modified carbon nonwoven fabric containing C_6_H_4_-CONH-CH_2_CH_2_-NH-DGR-Ac on the surface (**CF-3**).

**Figure 8 materials-14-03198-f008:**
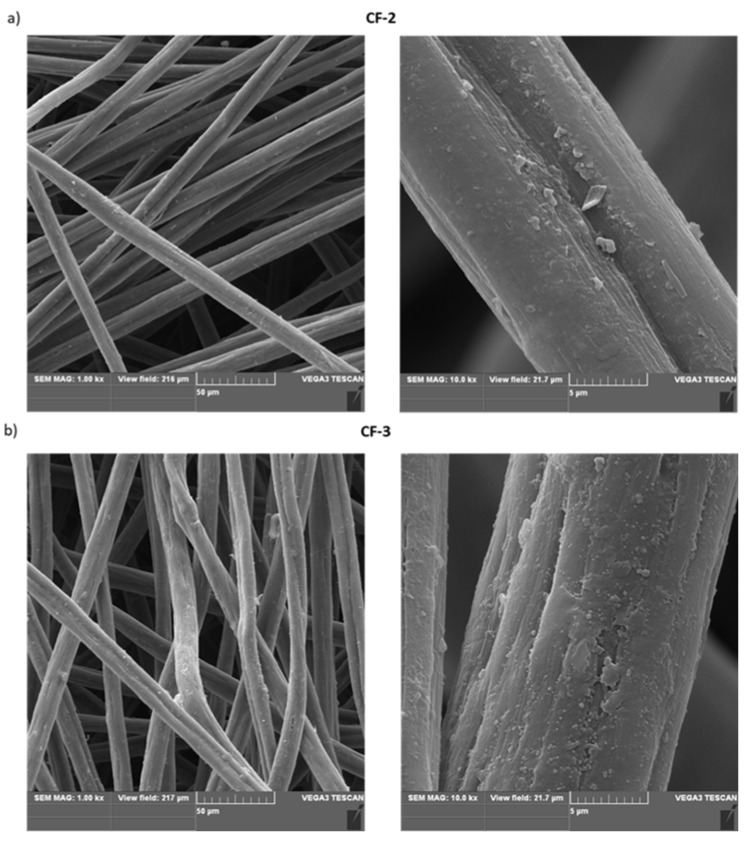
SEM images of: (**a**) modified carbon nonwoven fabric containing C_6_H_4_-CONH-CH_2_CH_2_-NH_2_ on the surface (**CF-2**); (**b**) modified carbon nonwoven fabric containing C_6_H_4_-CONH-CH_2_CH_2_-NH-DGR-Ac on the surface (**CF-3**).

**Figure 9 materials-14-03198-f009:**
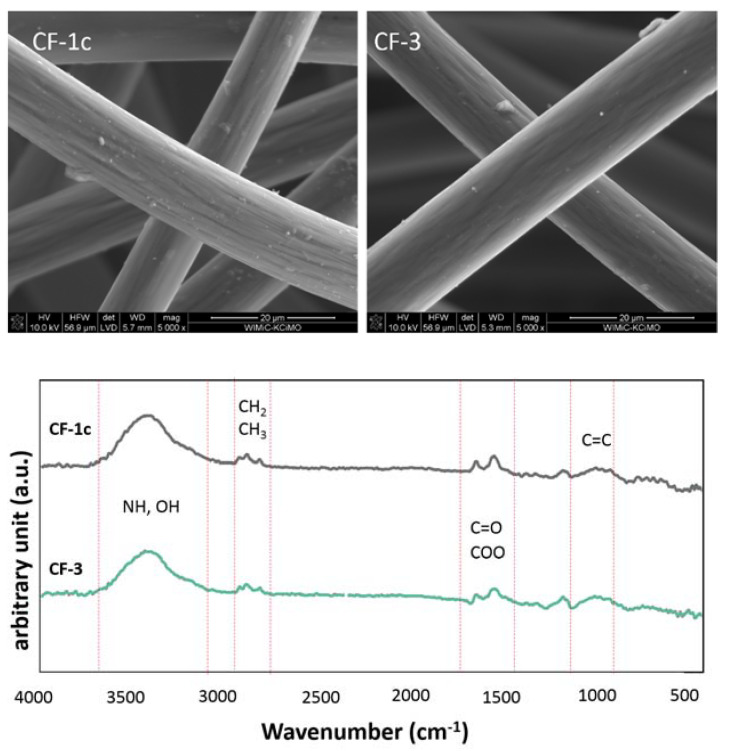
SEM images of modified carbon nonwoven fabric **CF-1c** and **CF-3** and FTIR-DRIFT spectrum of **CF-1c** and **CF-3**.

**Table 1 materials-14-03198-t001:** Characterization: fiber diameter, nonwoven porosity, and wettability of nonwovens obtained from PAN fibers, stabilized precursor (oxyPAN), and carbon nonwovens obtained after carbonization process (CF).

	PAN	oxyPAN	CF
Fiber diameter ± SD, µm	14.3 ± 1.0	9.7 ± 0.4	9.2 ± 0.2
Nonwoven retention, %	85.4 ± 3.0	75.9 ± 3.0	64.6 ± 3.0
Nonwoven porosity, %	87.8 ± 3.0	92.8 ± 3.0	95.6 ± 3.1
Nonwoven wettability	56.4 ± 3.1	66.7 ± 2.7	86.9 ± 3.4

**Table 2 materials-14-03198-t002:** Summary of physicochemical measurements: contact angle (CA) and surface free energy (SFE) for carbon nonwovens unmodified (CF) and chemically modified (**CF-1c**, **CF-3**).

Sample	Water CA, *θ* (deg)	Diiodomethane CA, *θ* (deg)	Surface Free Energy, SFE (mN/m)	Polar Component SFE, ɣp (mN/m)	Dispersive Component SFE, ɣd (mN/m)
CF	86.9 ± 2.8	43.3 ± 2.9	34.5 ± 2.4	12.9 ± 0.9	21.6 ± 1.5
**CF-1c**	68.4 ± 4.5	62.3 ± 4.4	30.8 ± 3.1	22.6 ± 2.5	8.2 ± 0.6
**CF-3**	66.9 ± 4.2	64.1 ± 4.5	31.1 ± 3.5	24.1 ± 1.6	7.0 ± 1.9

## Data Availability

Data sharing is not applicable to this article.

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
