# Peer review of "Chemical Modification as a Method of Improving Biocompatibility of Carbon Nonwovens"

_materials, 2021, doi:10.3390/ma14123198_

Round 1

Reviewer 1 Report

Authors of the current manuscript report the chemical modification of carbon nonwoven fabric spun from a polyacrylonitrile polymer solution. The nonwovens were treated chemically with aryl diazonium compounds in solution, which provided the direct linkage between the carbon substrates and chemical modifiers, and they were finally coupled with a peptide.

The surface modification protocol is well established in carbon nanotubes and has been used for similar purposes. The distinctive point of this manuscript is the adoption of carbon nonwovens, which may not be sufficiently intriguing in themselves without results of their performance in the suggested applications.

It would be desired that the authors show unique advantages of their nonwoven-based materials compared to related carbon substrates, for example, higher loading capacity of biomaterials or improved functions.

Author Response

Dear Referees, Dear Editor

I appreciate very much the comments given by You and Reviewer. In response to suggestions the following revision was introduced (marked in text by yellow background)

All Reviewers’ comments were carefully analyzed and necessary modifications in the manuscript were made.

All comments regarding missing elements of the publication and graphical material were taken into account and introduced to the manuscript

Reviewer 1

Authors of the current manuscript report the chemical modification of carbon nonwoven fabric spun from a polyacrylonitrile polymer solution. The nonwovens were treated chemically with aryl diazonium compounds in solution, which provided the direct linkage between the carbon substrates and chemical modifiers, and they were finally coupled with a peptide.

The surface modification protocol is well established in carbon nanotubes and has been used for similar purposes. The distinctive point of this manuscript is the adoption of carbon nonwovens, which may not be sufficiently intriguing in themselves without results of their performance in the suggested applications.

It would be desired that the authors show unique advantages of their nonwoven-based materials compared to related carbon substrates, for example, higher loading capacity of biomaterials or improved functions.

Thank you very much for your valuable comment. In fact, the method of functionalization of carbon nanomaterials containing Csp2 bonds that we used is widely known and used. Importantly, in our earlier work, we showed that it is even possible to use olefinated materials (content of Csp2 bonds significantly lower compared to carbon nanotubes or other Csp2 carbon materials) based on nanodiamonds for this type of functionalisation. In the case of carbon fibers formed from the PAN precursor and in the subsequent processing step into nonwovens, the properties of these materials (from the point of view of hybridization of carbon atoms) are similar to those of Csp2 materials. However, carbon fibers or nonwovens made of them differ significantly in physical and chemical properties from classic carbon materials (e.g. carbon nanotubes), where the functionalization method with the use of in situ generated diazonium salts is used for modification. In the case of fibers/nonwovens, the surface area, and thus the effectiveness of modification, is significantly lower than for carbon nanomaterials. Moreover, carbon fibers made of PAN in their structure contain additional structural elements derived from pyridine derivatives, which significantly distinguishes them from classic aromatic materials.

For this reason, we decided that it is necessary to carry out research aimed at checking the applicability of the method using in situ generated diazonium salts to modify the surface of carbon nonwovens derived from PAN. In addition, we expected that the introduction of polar groups or peptides on the surface of hydrophobic carbon material should affect cell adhesion due to a change in the polarity of the material's surface, and thus enable modulation of the material's ability to interact with cells. The results of the conducted research allowed us to demonstrate that the materials modified in this way are characterized by significantly higher wettability compared to unmodified materials. The results of these tests are presented in Supporting Material. This information was added to the manuscript.

Reviewer 2 Report

-RGD should be defined in the introduction

-The authors should provide references where clear evidence of physico-chemical parameters such as wettability is shown to affect cell adhesion, see, for instance, Yongabi et al. QCM-D study of time-resolved cell adhesion and detachment: Effect of surface free energy on eukaryotes and prokaryotes, ACS Applied Materials and Interfaces 12, 18258 (2020)

-The authors should clarify if they used sessile drop, how was the water contact angle of these fibers measured (size of the drop, fibers placed on a support)?

-Which kind of instrument is VEGA TESCAN? SEM? It is not indicated in the Materials section

- No explanation on how porosity percentage is obtained is given in the manuscript

-Table 1: Revise uncertainties and number of significant digits displayed!

- The authors mention: ‘It is essential for obtaining the appropriate mechanical properties of carbon nonwovens.’ How do they quantify these properties (young modulus?)?

-The correlation between the contact angles obtained (wettability) and the surface modification characterized by FTIR and SEM is missing. Did the authors consider measuring the surface energy of each system in order to decouple the polar and dispersive components? An example is provided in the reference suggested.

Author Response

Dear Referees, Dear Editor

I appreciate very much the comments given by You and Reviewer. In response to suggestions the following revision was introduced (marked in text by yellow background). All Reviewers’ comments were carefully analyzed and necessary modifications in the manuscript were made. All comments regarding missing elements of the publication and graphical material were taken into account and introduced to the manuscript

 Reviewer 2

  1. RGD should be defined in the introduction.

I sincerely apologize for missing RGD information in the manuscript. This has been corrected and added to the manuscript.

  1. The authors should provide references where clear evidence of physico-chemical parameters such as wettability is shown to affect cell adhesion, see, for instance, Yongabi et al. QCM-D study of time-resolved cell adhesion and detachment: Effect of surface free energy on eukaryotes and prokaryotes, ACS Applied Materials and Interfaces 12, 18258 (2020)

I absolutely agree with the valuable comment of the Reviewer. Apart from the aforementioned publication, other publications indicating the influence of the physico-chemical properties of scaffolds on the adhesion capacity of cells were added to the manuscript.

  1. The authors should clarify if they used sessile drop, how was the water contact angle of these fibers measured (size of the drop, fibers placed on a support)?

I sincerely apologize for missing information about test water contact angle in the manuscript. This has been corrected and added to the manuscript.

  1. Which kind of instrument is VEGA TESCAN? SEM? It is not indicated in the Materials section.

We added the SEM information in the manuscript.

  1. No explanation on how porosity percentage is obtained is given in the manuscript.

We are corrected information about porosity test.

  1. Table 1: Revise uncertainties and number of significant digits displayed!

The values was corrected in table 1.

  1. The authors mention: ‘It is essential for obtaining the appropriate mechanical properties of carbon nonwovens.’ How do they quantify these properties (young modulus?)?

We are added some reference in text, which we presented the changes of structure and mechanical properties of precursor and carbon fibers.

  1. The correlation between the contact angles obtained (wettability) and the surface modification characterized by FTIR and SEM is missing. Did the authors consider measuring the surface energy of each system in order to decouple the polar and dispersive components? An example is provided in the reference suggested.

We are added some comments about hydrophilic properties of modified carbon nonwovens in our manuscript.

Round 2

Reviewer 1 Report

The authors presented a justification why they chose carbon nonwovens and conducted research. The manuscript is satisfactorily revised.